# A Series of Field-Induced Single-Ion Magnets Based on the Seven-Coordinate Co(II) Complexes with the Pentadentate ($N_3O_2$) H$_2$dapsc Ligand

**Vyacheslav A. Kopotkov** [1,*] , **Denis V. Korchagin** [1,*] , **Valentina D. Sasnovskaya** [1],
**Ildar F. Gilmutdinov** [2] **and Eduard B. Yagubskii** [1,*]

1   Institute of Problems of Chemical Physics, Russian Academy of Sciences, Semenov's av., 1,
    Chernogolovka 142432, Russia; sasnovskayavd@rambler.ru
2   Institute of Physics, Kazan Federal University, 16a Kremlyovskaya st., Kazan 420008, Russia;
    ildar.gilmutdinov@gmail.com
*   Correspondence: slavaoven@mail.ru (V.A.K.); korden@icp.ac.ru (D.V.K.); yagubski@icp.ac.ru (E.B.Y.);
    Tel.: +7-49-6522-1256 (V.A.K.); +7-49-6522-1531 (D.V.K.); +7-49-6522-1185 (E.B.Y.)

**Abstract:** A series of five new mononuclear pentagonal bipyramidal Co(II) complexes with the equatorial 2,6-diacetylpyridine bis(semicarbazone) ligand (H$_2$dapsc) and various axial pseudohalide ligands (SCN, SeCN, N(CN)$_2$, C(CN)$_3$, and N$_3$) was prepared and structurally characterized: [Co(H$_2$dapsc) (SCN)$_2$]·0.5C$_2$H$_5$OH (**1**), [Co(H$_2$dapsc) (SeCN)$_2$]·0.5C$_2$H$_5$OH (**2**), [Co(H$_2$dapsc) (N(CN)$_2$)$_2$]·2H$_2$O (**3**), [Co(H$_2$dapsc) (C(CN)$_3$)(H$_2$O)](NO$_3$)·1.16H$_2$O (**4**), and {[Co(H$_2$dapsc) (H$_2$O) (N$_3$)][Co(H$_2$dapsc)(N$_3$)$_2$]}N$_3$·4H$_2$O (**5**). The combined analyses of the experimental *DC* and *AC* magnetic data of the complexes (**1–5**) and two other earlier described those of this family [Co(H$_2$dapsc) (H$_2$O)$_2$)](NO$_3$)$_2$·2H$_2$O (**6**) and [Co(H$_2$dapsc)(Cl)(H$_2$O)]Cl·2H$_2$O (**7**), their theoretical description and the ab initio CASSCF/NEVPT2 calculations reveal large easy-plane magnetic anisotropies for all complexes ($D = + 35 − 40$ cm$^{-1}$). All complexes under consideration demonstrate slow magnetic relaxation with dominant Raman and direct spin–phonon processes at static magnetic field and so they belong to the class of field-induced single-ion magnets (*SIMs*).

**Keywords:** single ion magnets; seven-coordinate complexes; Co(II) complexes; ligand H$_2$dapsc; crystal structure; DC and AC magnetic properties

## 1. Introduction

Large uniaxial magnetic anisotropy has a crucial role in the enhancement of the blocking temperature of magnetization reversal in single molecular magnets (*SMMs*) [1–4]. One of the approaches to increase magnetic anisotropy is to use special ligands with a less-common coordination since the anisotropy depends on the coordination number of the metal centers [5,6]. The magnetic anisotropy can be improved by a rational ligand design [7–15]. The "ligand approach" has already led to a variety of mononuclear 3*d* metal ion—based *SMMs* (so-called single-ion magnets (*SIMs*)) with improved characteristics [6]. Among them, the two-coordinate complex of Fe(I) exhibited an effective spin-reversal barrier of $U_{eff} = 226$ cm$^{-1}$ [7] and recently reported that the linear two-coordinated Co(II) imido complex has the energy barrier of 413 cm$^{-1}$ [13], the largest yet observed for a 3*d*-based *SMM*. However, low coordinate complexes are not very stable which restricts their possible utility in different applications. At the same time, the experimental and theoretical studies of stable seven-coordinated metal centers with pentagonal bipyramidal geometry showed that such centers are highly promising as anisotropic spin carriers [16–25]. In this context, 3*d*-metallocomplexes with the pentadentate Schiff-base H$_2$dapsc ligand (H$_2$dapsc = 2,6-diacetylpyridine bis(semicarbazone), Figure 1) and its analogues

are of considerable interest. These complexes reveal the rare occurrence of pentagonal bipyramidal stereochemistry about a central metal ion, which results from pentacoordination of the nearly planar $H_2dapsc$ ($N_3O_2$), and two labile apical ligands ($H_2O$ and/or Cl, $NO_3$, and others) perpendicular to the equatorial pentagon plane [26–30]. Although the complexes of $H_2dapsc$ and its analogues have long been known, the study of their magnetic properties [17,19–21,31–34] and their use as building blocks for the preparation of magnetic polynuclear assemblies [21,35–38] are started only in last time. It has been shown that some of these Ni(II), Fe(II), and Co(II) mononuclear complexes with the $H_2dapbh$, $H_2dapsc$, and $H_4daps$ ligands (Figure 1) demonstrate slow magnetic relaxation. In contrast to the Ni(II) and Fe(II) complexes which reveal strong uniaxial magnetic anisotropy ($D < 0$) [20,21,31,35] characteristic of *SIMs*, the known Co(II) complexes with $H_2dapbh$ and $H_2daps$ ligands revealed a large easy-plane magnetic anisotropy ($D > 0$) [17,19,20,32–34]. Although *D*-parameter has a positive value which in accordance with the theory should not allow the *SIM* behavior, these Co(II) complexes show slow relaxation of magnetization in the presence of static magnetic field (so-called field-induced *SIMs*). Ruis, Luis, and co-workers gave the explanation of the presence of the field-induced slow magnetic relaxation for Kramer ions, such as Co(II), with prevailing easy-plane magnetic anisotropy [39]. The origin of this large non-uniaxial anisotropy is due to the mixing of the ground electronic state with the excited electronic states because of spin-orbit coupling.

**Figure 1.** Molecular structure of the pentadentate ligand $H_2dapsc$ (R = $NH_2$) and some its analogues: $H_2dapbh$ (R = $C_6H_5$), $H_2biph$ (R = $C_6H_4$-$C_6H_5$), and $H_4daps$ (R = 2-$OHC_6H_4$).

In this paper, we present the synthesis and crystal structures of the five new seven-coordination Co(II) complexes with $H_2dapsc$ equatorial and different pseudohalide axial ($SCN^-$, $SeCN^-$, $[N(CN)_2]^-$, $[C(CN)_3]^-$, $N_3^-$) ligands: $[Co(H_2dapsc)(SCN)_2]\cdot0.5C_2H_5OH$ (**1**), $[Co(H_2dapsc)(SeCN)_2]\cdot0.5C_2H_5OH$ (**2**), $[Co(H_2dapsc)(N(CN)_2)_2]\cdot2H_2O$ (**3**), $[Co(H_2dapsc)(C(CN)_3)(H_2O)](NO_3)\cdot1.16H_2O$ (**4**), and $\{[Co(H_2dapsc)(H_2O)(N_3)][Co(H_2dapsc)(N_3)_2]\}N_3\cdots4H_2O$ (**5**). Molecular and crystal structures of these complexes were investigated by single crystal X-ray diffraction method. The *DC* and *AC* magnetic properties of these complexes and two other described those of this family $[Co(H_2dapsc)(H_2O)_2)](NO_3)_2\cdot2H_2O$ [28] (**6**) and $[Co(H_2dapsc)(Cl)(H_2O)]Cl\cdot2H_2O$ [29] (**7**) were studied. The detailed theoretical analysis of magnetic properties was provided. The effect of modification of axial ligands on magnetic anisotropy was traced.

## 2. Results and Discussion

### 2.1. Synthesis and Characterization

The Co(II) complexes with $H_2dapsc$ ligand and pseudohalide anions were synthesized using the following two approaches: (*A*) the substitution of the terminal ligands ($H_2O$) in the starting $Co^{2+}$ complex with $H_2dapsc$ ligand $[Co(H_2dapsc)(H_2O)_2](NO_3)_2$ [28,29] for pseudohalide anions; and (*B*) the reaction between $Co^{2+}$ nitrate, $H_2dapsc$ ligand, and a precursor of the pseudohalide anions, Scheme 1.

| CatY | Y' | Y" | Z | Complex |
|------|------|------|------|---------|
| KSCN | SCN | SCN | - | (**1**) |
| KSeCN | SeCN | SeCN | - | (**2**) |
| NaN(CN)$_2$ | N(CN)$_2$ | N(CN)$_2$ | - | (**3**) |
| KC(CN)$_3$ | C(CN)$_3$ | H$_2$O | NO$_3$ | (**4**) |

Method *A*

(H$_2$dapsc)

Method *B*

**Scheme 1.** Synthesis of complexes **1**–**5**. Methods *A* and *B* are described in the text.

The IR spectra of the obtained complexes are quite similar. The presence of the H$_2$dapsc ligand was confirmed by the following characteristic vibrations: $\nu$(N–H) and $\nu$(C=N) vibrations of the amino and imino groups were located in the 3150–3350 cm$^{-1}$ and 1650–1700 cm$^{-1}$ region, respectively. Bands for $\nu$(C≡N) and $\nu$(–N=N$^+$=N$^-$) stretching vibrations of the pseudohalide anions in **1**–**5** were observed in the 2050–2200 cm$^{-1}$ region.

## 2.2. Description of the Structure

Compounds **1** and **2** are isostructural. They crystallize in the monoclinic space group *P2$_1$/c* with a half solvate EtOH molecule per one molecule of complex (Figure 2a). Solvate EtOH molecule occupies special positions (center of symmetry) with 1/2 occupancies. Compound **3** crystallizes in the triclinic space group *P-1* with a two solvate water molecules per one molecule of complex (Figure 2b). In contradistinction to neutral **1**–**3** complexes, **4** is cationic. It crystallizes in the monoclinic space group *P2$_1$/c* with a one nitrate-anion and two solvate water molecules, one of them (O2w) has site occupation factor ~0.16 (Figure 2c). Complex **4** contains apical ligands of different nature: Tricyanomethanide-anion ([C(CN)$_3$]$^-$ = tcm) and H$_2$O, whereas in the complexes **1**–**3** the apical ligands are the same SCN$^-$, SeCN$^-$, [N(CN)$_2$]$^-$, respectively.

It is surprising that crystals of **5** contain two complexes in the same lattice linked by hydrogen bonds (see below): Neutral [Co(H$_2$dapsc)(N$_3$)$_2$] and cationic [Co(H$_2$dapsc)(N$_3$)(H$_2$O)]$^+$, Figure 2d. Compound **5** crystallizes in the monoclinic space group *C2/c* with a two solvate water molecules per one molecule of complexes (Figure 2d). Isolated azide anion in special position plays the role of counterion. The structure of both complexes is very close so that we could solve and refine the crystal structure with one molecule of the complex in asymmetric unit (in order to minimize number of parameters).

Crystal structures **6** and **7** have been previously studied (Figure 2e,f) and described [28,29].

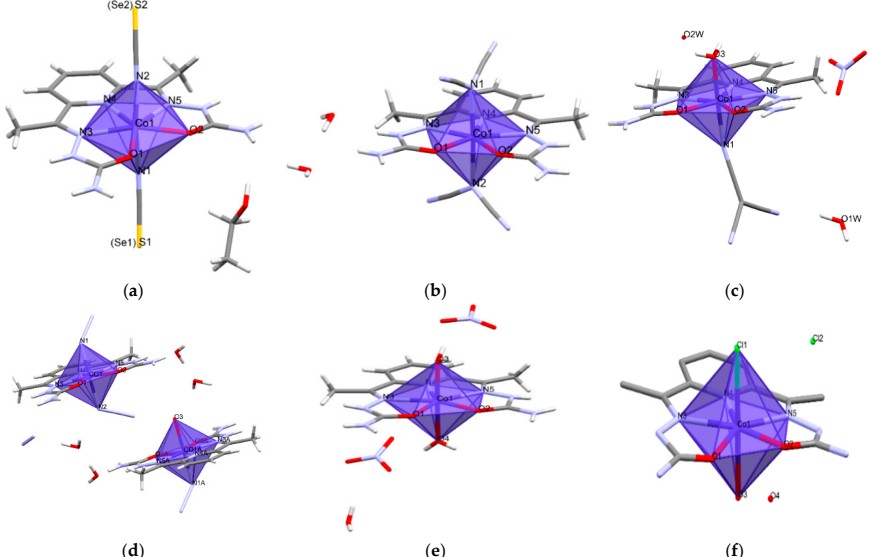

**Figure 2.** The molecular structures of **1** (**2**) (**a**), **3** (**b**), **4** (**c**), **5** (**d**), **6** (**e**), and **7** (**f**). Disordering of solvate molecules is omitted for clarity (see text).

In all complexes the central Co(II) ion has a pentagonal bipyramidal coordination environment formed by five equatorial $N_3O_2$ (N3, N4, N5, O1, and O2) atoms from the $H_2$dapsc ligand and two axial N(O) atoms (N1, N2(O3)) from the axial $NCS^-$ $NCSe^-$, $N(CN)_2^-$, tcm, $N_3^-$, or $H_2O$ ligands. In previously studied compounds **6** [28] and **7** [29] axial ligands are two water molecules and water molecule and chloride-anion, respectively. In compound **3**, the relatively rare monodentate coordination of the $N(CN)_2^-$ ligand is realized. Usually (>75% of cases), dicyanamide anion acts as a polydentate bridging ligand [40]. Interestingly, the monodentate type of coordination is more characteristic of the tcm anion than on the dicyanamide anion. The number of examples of mono- and polydentate coordination of tcm ligands in Cambridge Structural Database [40] are almost the same. The axial bond angle N(1)-Co(1)-N(2) for **1** (or **2**) and **3** are ~174 and ~177°, respectively. In complexes **4** and **5** it has been found the smallest and the largest axial bond angles N(1)-Co(1)-O(3) 171.21(9) and 178.51(9), respectively. The axial Co-N(O) bond lengths in the complexes **1**, **2**, and **4** are slightly shorter than the equatorial Co-N(O) bond distances while in **3** Co-$N_{ax}$ bonds are similar to Co-(N)$O_{eq}$ (Table S2). At the same time, the Co-$O_{eq}$ bond lengths in **3** are slightly shorter than in other complexes. In complexes **1** and **2** the equatorial N-Co-O(N) bond angles are in the more large range 69.8–77.8° than in **3** (70.2–75.6°), that together with approximately equal bond lengths in the coordination polyhedron $CoN_5O_2$ of **3** indicates to a more distorted pentagonal bipyramidal geometry of Co(II) complexes **1** and **2**. Indeed, the SHAPE software [41,42] gave the deviation parameters of 0.058 for **3** and 0.312 (0.314) for **1** (**2**), Table S1, which in the case of **1** (**2**) is larger from zero of the ideal $D_{5h}$ symmetry that confirms more distorted coordination polyhedra of **1** and **2** complexes. The equatorial N-Co-O(N) bond angles in **3** (70.2–75.6°) and **4** (70.1–76.2°) are similar but the strong distortion of axial bond angle N(1)-Co(1)-O(3) and the difference of axial and equatorial bond lengths (see Table S2) in **4** leads to the relatively large deviation parameter 0.294 as in the case of complexes **1** and **2**. In spite of similarity azide and isothiocyanate anions, Co(II) coordination environment in **5** is closer to **3** than **1** or **2** (see Table S2). Also the SHAPE program (Table S1) gave the deviation parameters of 0.105 and 0.072 for neutral [Co($H_2$dapsc)($N_3$)$_2$] and cationic [Co($H_2$dapsc)($N_3$)($H_2O$)]$^+$ in **5** that shows less distorted coordination polyhedra of these complexes than in the cases of **1** and **2**.

Figure S1a,b shows the fragments of crystal structures of **1** and **2**. Crystal structures are stabilized by the number of intermolecular hydrogen bonds (H-bonds) between complexes itself, and between complexes and solvate EtOH molecules (Table S3). The Co(II) ions in crystal packing are not well isolated with the closest intermolecular Co···Co separations being 6.96 (**1**) and 7.01Å (**2**). In case of **3**, the closest intermolecular Co···Co separations are somewhat larger (7.48Å), apparently due to the

larger size of the apical ligands or the presence of a larger number of solvate water molecules in the crystal structure.

The presence of not only solvate water molecules but also isolated counterions ($NO_3^-$ and $N_3^-$) in the crystal structures of **4** and **5** gave the largest from shortest intermolecular Co···Co separations (7.66 and 7.73 Å, respectively) among all structures under consideration. Crystal structures of **3**−**5** is also stabilized by the large number of H-bonds with the amino and imino groups of $H_2$dapsc ligands, solvate water molecules, and $NO_3^-$ anions (Figures S1c,d and S2, Table S3). It should be noted that the strong distorted axial bond angle N(1)-Co(1)-O(3) in **4** complex can be the result of stacking interactions of tcm ligands of adjacent molecules (Figure S3) and/or H-bonds involving a water ligand.

As it has been already noted in the works [28,29], the crystal structures of **6** and **7** are stabilized by an extensive network of hydrogen bonds involving the complex cations, the nitrate or chloride anions, and the solvent water molecules (Figure S1e,f). In crystal packings of **6** and **7** Co(II) ions are not well isolated with the closest intermolecular Co···Co separations being 6.65 and 6.75 Å, respectively, these values are the two smallest values from all compound under consideration.

## 2.3. Magnetic Properties

### 2.3.1. Static Magnetic Measurements

The temperature dependencies of magnetic susceptibility for complexes **1**–**7** were performed in the temperature range of 2.0–300 K under a 5000 Oe *DC* field. The shapes of the $\chi_M T$ versus $T$ curves of all complexes are similar. At room temperature, $\chi_M T$ products are in the range of 2.36–2.61 cm$^3$·K·mol (Figure 3). These values are higher than the spin only value for Co(II) with $S = 3/2$ (1.875 cm$^3$ K mol$^{-1}$) due to an orbital contribution to the magnetic moment. Upon cooling, $\chi_M T$ remains almost constant in the temperature range from 300 to 60 K after which point it starts to decrease for all the compounds and reaches the values of $\approx$1.4–1.6 cm$^3$·K·mol$^{-1}$ at 2 K (Figure 3). The room-temperature $\chi_M T$ value 5.14 cm$^3$·K·mol$^{-1}$ for **5** is in good agreement with the paramagnetic response of the two magnetically non-interacting Co(II) ions with $S = 3/2$ (Figure 3e). The decrease of the $\chi_M T$ at low temperatures is due to the intrinsic magnetic anisotropy of the Co(II) centers [43]. Field dependences of magnetization for all complexes at 2 K (insets on Figure 3) saturate at values of around 2.15–2.28 $N_A \mu_B$ that is significantly lower than the value of $3 N_A \mu_B$, corresponding to the pure spin $S = 3/2$ ground state with $g = 2$. This fact indicates the presence of considerable magnetic anisotropy in the complexes.

**Table 1.** The CASSCF/NEVPT2 quantum chemical calculated and fitted magnetic parameters for **1**–**7**.

| | 1 | 2 | 3 | 4 | 5a | 5b | 6 | 7 |
|---|---|---|---|---|---|---|---|---|
| **ZFS and *g* values based on CASSCF/NEVPT2 calculations with CAS(7,5)** | | | | | | | | |
| $D$ (cm$^{-1}$) | 38.02 | 37.73 | 37.51 | 37.49 | 39.93 | 39.58 | 38.99 | 38.94 |
| $E/D$ | 0.0227 | 0.0213 | 0.0065 | 0.0093 | 0.0162 | 0.0143 | 0.0143 | 0.0133 |
| $g_x$ | 2.3442 | 2.3366 | 2.3220 | 2.3206 | 2.3590 | 2.3490 | 2.3284 | 2.3511 |
| $g_y$ | 2.3641 | 2.3554 | 2.3269 | 2.3286 | 2.3736 | 2.3617 | 2.3559 | 2.3625 |
| $g_z$ | 2.0049 | 2.0023 | 1.9936 | 1.9943 | 1.9971 | 1.9949 | 1.9925 | 2.0009 |
| $g_{iso}$ | 2.2377 | 2.2314 | 2.2148 | 2.2145 | 2.2432 | 2.2352 | 2.2256 | 2.2382 |
| **Analysis of the experimental magnetic data** | | | | | | | | |
| $D$ (cm$^{-1}$) | 35.6 | 38.20 | 35.3 | 33.60 | 40.4 | | 38.02 | 35.61 |
| $E/D$ | 0.17 | 0.00 | 0.101 | 0.149 | - | | 0.018 | 0.16 |
| $g_{x,y}/g_z$ | 2.29/2.14 | 2.36/1.90 | 2.28/2.13 | 2.26/2.00 | 2.48/2.00 | | 2.28/2.16 | 2.45/2.11 |
| $g_{iso}$ | 2.24 | 2.26 | 2.23 | 2.18 | 2.33 | | 2.24 | 2.34 |
| $\chi_{TIP}$ | - | $1.0 \times 10^{-4}$ | $1.0 \times 10^{-4}$ | $5.0 \times 10^{-4}$ | - | | $5.0 \times 10^{-4}$ | - |

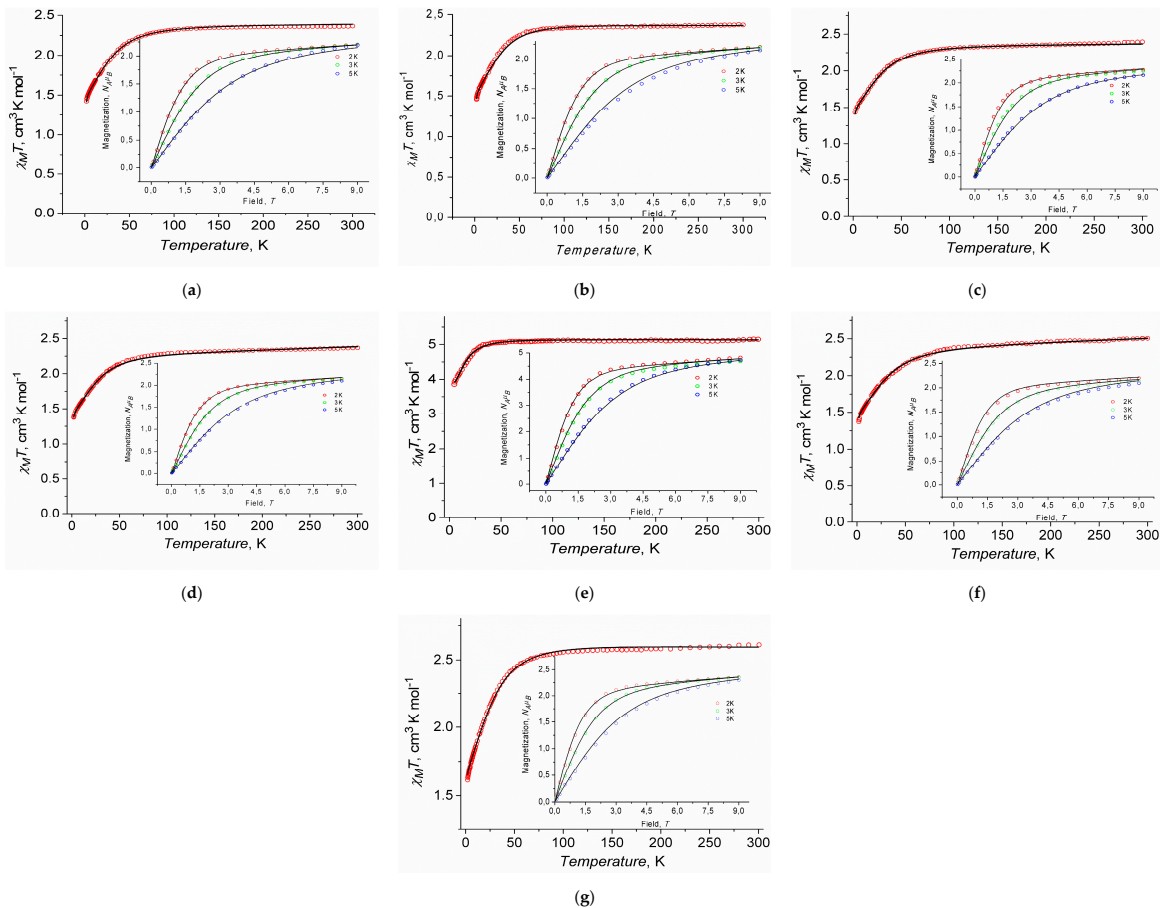

**Figure 3.** Temperature dependence of $\chi_M T$ obtained at 0.5 T for **1** (**a**), **2**(**b**), **3** (**c**), **4** (**d**), **5** (**e**), **6** (**f**), and **7** (**g**). The insets: Magnetization versus magnetic field measured at *T*= 2, 3, and 5 K for these complexes. The empty circles represent the experimental data and black solid lines represent the fitted using Equation (1) with the parameters listed in Table 1.

To describe the *DC* magnetic properties, we use the following zero-field splitting (*ZFS*) spin Hamiltonian:

$$\hat{H} = D\left[\hat{S}_Z^2 - \frac{1}{3}S(S+1)\right] + E\left(\hat{S}_X^2 - \hat{S}_Y^2\right) + \mu_B\left(B_X g_X \hat{S}_X + B_Y g_Y \hat{S}_Y + B_Z g_Z \hat{S}_Z\right) \tag{1}$$

here *S* = 3/2 is the spin of the high-spin Co(II) ion, *D* and *E* are axial and rhombic *ZFS* parameters, and $g_\alpha$ ($\alpha$ = X, Y, Z) are the principle values of the *g*-tensor. The set of the best-fit parameters for the observed temperature dependences of $\chi_M T$ (Figure 3) and field dependences of magnetization at different temperatures (Figure 3) is presented in the Table 1. These parameters agree well with the calculated by SA-CASSCF/NEVPT2 methods set of parameters (Table 1). The complexes exhibit positive axial magnetic anisotropy (*D*) with non-zero rhombicity parameter (Table 1). The sign and the magnitudes of *D* for **1**–**7** are in the range of previously reported those for the seven-coordinate pentagonal bipyramidal Co(II) complexes [24,25,33,34].

### 2.3.2. Quantum Chemical Calculations

The first excited quartet term lies above 3047 cm$^{-1}$ is well separated from the excited ones for all complexes (Figure 4a). This is also reflected in the ligand field multiplets (Kramers doublets) calculated with the spin–orbit coupling, where in all cases, two lowest Kramers doublets can be described with the ZFS formalism using *S* = 3/2, because the third doublet is also well separated from the ground state by an energy gap more than 2450 cm$^{-1}$ (Figure 4b).

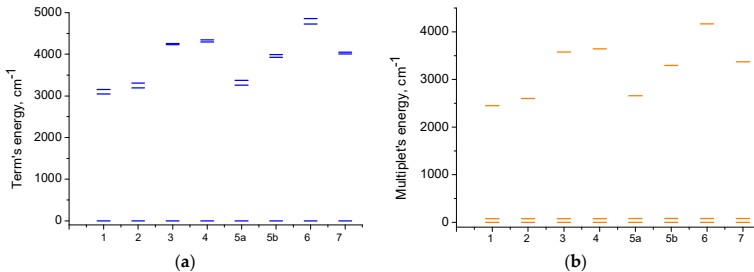

(a)  (b)

**Figure 4.** The lowest energy levels from SA-CASSCF(7,5)/NEVPT2 calculations for **1**–**7** (the ground and excited quartet states (**a**) and the lowest ligand field multiplets/Kramers doublets result from the spin–orbit coupling (**b**).

The values of the *ZFS* parameters *D* and *E/D* as well as the g values calculated by introducing the spin-orbit coupling (*SOC*) operators, extracted with the aid of the effective Hamiltonian approach, are listed in Table 1, orientation of the corresponding magnetic axes are given in Figure 5.

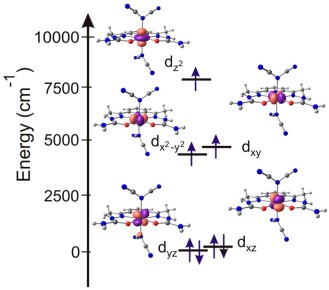

**Figure 5.** *d*-AO diagram according to ab initio ligand field theory (AILFT) analysis for **3**.

The results of the calculations are in good agreement in both magnitude and sign with the experimental data for all complexes. The analysis of the *g*-tensor components shows the closeness of $g_x$ and $g_y$ values that confirms the correctness of the choice of $g_x = g_y$ in analysis of magnetic data. The rhombicity parameter $E/D = |D_x - D_y|/2$ is small enough this fact correlates with the proximity of the $g_x$ and $g_y$ values.

Splitting of the d-orbitals (Figure 5) has been calculated and analyzed within the ab initio ligand field theory (AILFT). Two doubly occupied low-lying states ($d_{yz}$ and $d_{xz}$) are close in energy and are well separated from single occupied one-electron states (see Table 2). The analysis of individual excited states contributions to the total *D* value (Table 3) shows, that the main contribution with positive sign goes from the third and fourth quartet excited states, also major contribution from doublet state goes from various excited doublet states (these three or four states give about 100% of the calculated total *D* value). It should be noted that **1** and **2** complexes are different from other by the nature of *SOC* contributions of doublet excited states (Table 3).

**Table 2.** Relative energies (cm$^{-1}$) of ligand field one-electron states (in the basis of d-AOs) from AILFT analysis (SA-CASSCF(7,5)/NEVPT2).

| d-AO | 1 | 2 | 3 | 4 | 5a | 5b | 6 | 7 |
|------|-----|-----|-----|-----|-----|-----|-----|-----|
| $d_{yz}$ | 0.0 | 0.0 | 0.0 | 0.0 | 0.0 | 0.0 | 0.0 | 0.0 |
| $d_{xz}$ | 253.2 | 274.5 | 119.3 | 326.3 | 513.4 | 254.8 | 330.2 | 339.0 |
| $d_{x2-y2}$ | 2888.0 | 3026.5 | 4437.7 | 4652.2 | 3779.5 | 4244.3 | 5142.2 | 4371.6 |
| $d_{xy}$ | 3499.5 | 3688.2 | 4741.3 | 4770.7 | 4019.6 | 4445.7 | 5241.4 | 4531.5 |
| $d_{z2}$ | 8044.8 | 8141.7 | 7885.8 | 7817.5 | 7742.2 | 7005.5 | 6019.4 | 6732.7 |

**Table 3.** Main contributions of the excited states to total *D* values (cm$^{-1}$).

| Excited State | | Contributions to Total *D*, cm$^{-1}$ | | | | | | | |
|---|---|---|---|---|---|---|---|---|---|
| No. | Mult | 1 | 2 | 3 | 4 | 5a | 5b | 6 | 7 |
| 3 | 4 | 17.26 | 16.79 | 15.03 | 15.65 | 17.45 | 17.09 | 17.03 | 16.62 |
| 4 | 4 | 15.27 | 14.79 | 15.14 | 14.90 | 16.26 | 16.30 | 14.20 | 15.74 |
| 5 | 2 | - | - | 13.28 | 13.51 | 11.65 | 13.15 | 13.63 | 12.82 |
| 6 | 2 | - | 4.56 | - | | | | | |
| 7 | 2 | | | −1.55 | −1.51 | −1.33 | −1.61 | −1.58 | −1.53 |
| 8 | 2 | 6.48 | 4.71 | −1.59 | −1.55 | −1.59 | −1.57 | −1.51 | −1.63 |

### 2.3.3. AC Susceptibility Data

To probe the possible *SIM* properties of compounds **1**–**7** alternating current susceptibilities as a function of frequency were measured at different temperatures for each compound in zero and non-zero *DC* field. In the absence of *DC* field, no obvious frequency dependence in the in-phase ($\chi'$) and out-of–phase ($\chi''$) can be observed for all compounds. The *DC* field-dependencies of *AC* susceptibility were recorded under fields of 0–3000 Oe at 2.0 K and two frequencies (100 Hz, 1000 Hz) in search of an optimal field for the suppression of quantum tunneling of magnetization (*QTM*), Figure S4. When applying a *DC* field of 1000–2000 Oe, a maximum of $\chi''$ was observed for all complexes, Figure S4. To probe the relaxation behavior of complexes **1**–**7**, the frequency dependence of *AC* susceptibility was studied in the presence of the optimal static *DC* field for each complex at different temperatures (Figure 6). Both the $\chi'$ and $\chi''$ susceptibilities show frequency-dependent signals indicating slow relaxation of magnetization. These results clearly indicate that the mononuclear complexes **1**–**7** are field-induced *SIMs*. The $\chi''$ peaks for different frequencies appear one after another and the frequency dependence data show the clear and steady shift of the peaks towards higher frequencies with increasing temperature that is characteristic of *SMMs*. In contrast to complexes **1**–**3** and **5**–**7**, in which one set of maxima is observed on the frequency dependencies of $\chi''$ (Figure 6), in the complex **4** at T < 5 K, there is a second high frequency set of maxima (Figure 6), which indicates the existence of the two-step relaxation processes in **4** [44,45]. Only the main low frequency peaks in **4** will be discussed here. More detailed information on the effects of the applied *DC* field on the slow magnetic relaxation processes and the reasons for observing two or even three relaxation processes in Co(II) complexes can be obtained from the works of Boča and co-workers [46–49]. The obtained data for the *AC* susceptibility were fitted in *cc-fit* program [50] by using the one- and two-component generalized Debye model for the complexes **1**–**3**, **5**–**7** and **4**, respectively [45,51], which gave the values and distribution of the relaxation time ($\tau$ and $\alpha$; Tables S4–S10). The $\alpha$ values are in the range 0.02−0.34 for all complexes suggesting the narrow distribution of relaxation time. The best representation of the obtained parameters is Cole-Cole plots (Figure S5). The simulations using these parameter sets are in a good agreement with the experimental data. For all complexes, the relaxation times ($\tau$) were plotted versus $T^{-1}$, giving Arrhenius–like curves (Figure S5), which were successfully fitted in whole temperature range with following equation:

$$\tau^{-1} = CT^n + A(H_{DC})^2 T \tag{2}$$

where the first and second terms describe Raman and direct spin–phonon processes, respectively. The direct one-phonon process is dominating at low temperatures, while the contribution of the two-phonon Raman process becomes important with increasing temperature. The best-fit parameters for **1**–**7** are summarized in Table 4. The Orbach process was not included in the relaxation time $\tau$ approximation for these complexes because there is no any real energy barrier between spin states in the Co(II) complexes with *D* > 0 [39,52,53]. As regards the process of fast temperature independent relaxation via *QTM*, this process is partially or completely suppressed in the presence of a small external *DC* field [54].

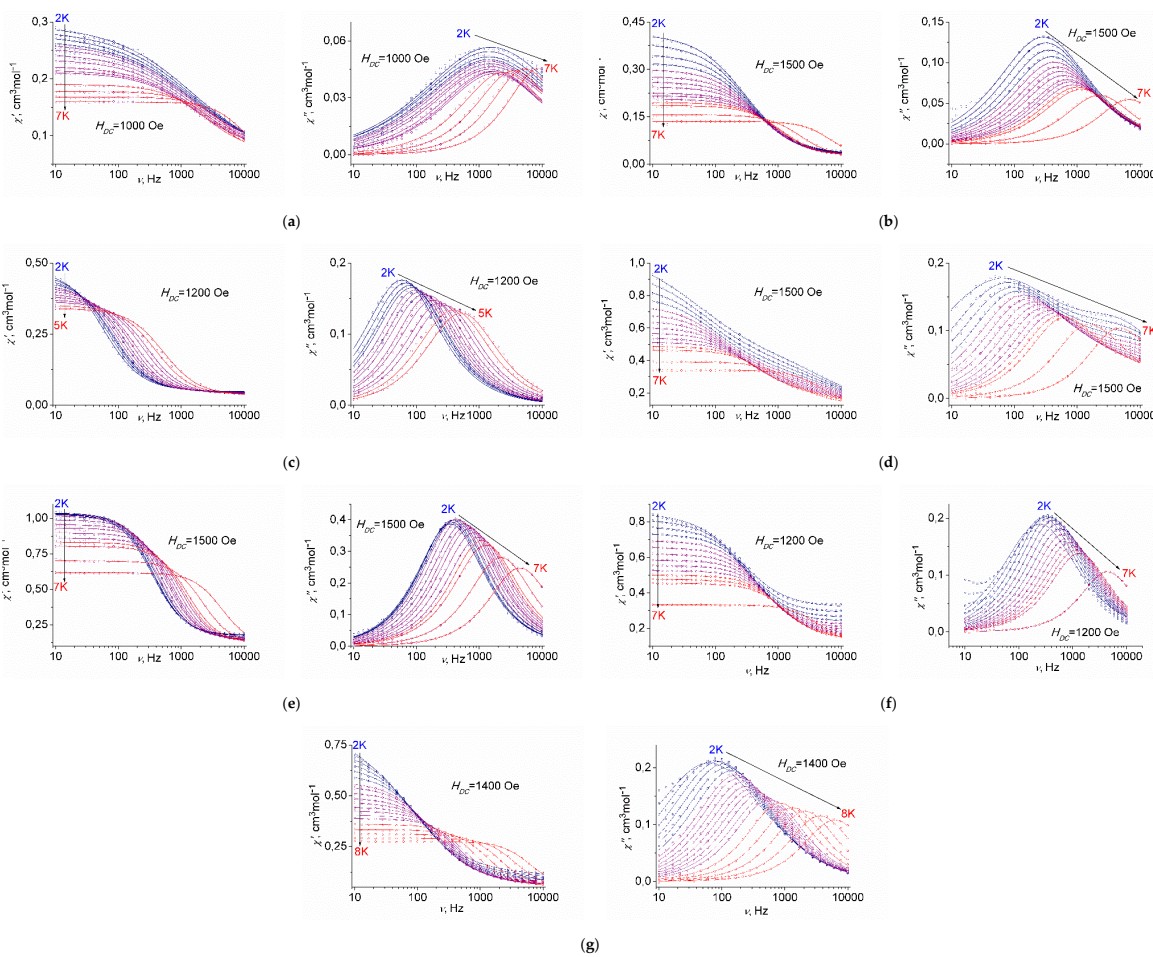

**Figure 6.** Frequency-dependence of in-phase ($\chi'$) and out-of-phase ($\chi''$) *AC* susceptibility in the temperature range of 2–8 K for **1** (**a**), **2**(**b**), **3** (**c**), **4** (**d**), **5** (**e**), **6** (**f**), and **7** (**g**) under a *DC* fields.

**Table 4.** Parameters for the magnetic relaxation of complexes **1**–**7** obtained by fit of the experimental relaxation times to Equation **2**.

|  | 1 | 2 | 3 | 4 | 5 | 6 | 7 |
|---|---|---|---|---|---|---|---|
| $n$ | 9 (*Fixed\**) | 7.4(3) | 7.3(2) | 5.6(3) | 5.8(4) | 4.2(2) | 9 (*Fixed\**) |
| $A$, $Oe^{-2}s^{-1}K^{-1}$ | $1.03(2) \times 10^{-3}$ | $4.10(9) \times 10^{-4}$ | $1.29(2) \times 10^{-4}$ | $7.7(6) \times 10^{-5}$ | $2.9(1) \times 10^{-4}$ | $2.9(1) \times 10^{-4}$ | $1.11(3) \times 10^{-4}$ |
| $C$, $s^{-1}K^{-n}$ | $1.06(6) \times 10^{-3}$ | 0.02(1) | 0.017(6) | 0.4(2) | 0.26(6) | 0.26(6) | $2.43(5) \times 10^{-4}$ |

\* Fixed $n$ = 9, since during approximation the parameter n exceeded this value, which is characteristic for the Raman process ($\sim CT^9$).

## 3. Materials and Methods

General remarks: The ligand $H_2$dapsc and complexes $[Co(H_2dapsc)(H_2O)_2](NO_3)_2 \cdot 2H_2O$ (**6**), $[Co(H_2dapsc)(H_2O)(Cl)]Cl\ 2H_2O$ (**7**) were prepared by a methods described in references [27–29]. All other chemicals were used as supplied from Aldrich.

### 3.1. Synthesis

$[Co(H_2dapsc)(SCN)_2] \cdot 0.5C_2H_5OH$ (**1**). *Method A.* The complex $[Co(H_2dapsc)(H_2O)_2](NO_3) \cdot 2H_2O$ [28] (0.146 g, 0.5 mmol) was dissolved in ethanol (15 mL). The solution was heated to 50 °C and stirred for 10–15 min, and then KSCN (97 mg, 1 mmol) in methanol (5 mL) was added. The resulting clear solution was filtered and cooled. The solution was slowly evaporated at 20 °C for one to three days. The precipitated red brown crystals were filtered off, washed with a small amount of water, and dried in

vacuo. For **1** yield: 40%. Anal. calc. for $CoC_{15}H_{21}N_9O_3S_2$ (498.46): C, 36.11; H, 4.21; N, 25.29%. Found: C, 35.81; H, 4.63; N, 25.03%. IR data (cm$^{-1}$): $\nu$(N–H) = 3176, 3334; $\nu$(C≡N) = 2075; $\nu$ (C=N) = 1652.

Complexes [Co(H$_2$dapsc)(SeCN)$_2$]·0.5C$_2$H$_5$OH (**2**), [Co(H$_2$dapsc)(N(CN)$_2$)$_2$]·2H$_2$O (**3**) and [Co(H$_2$dapsc)(C(CN)$_3$)(H$_2$O)](NO$_3$)·2H$_2$O (**4**) were obtained similarly to complex (**1**) using KSeCN, NaN(CN)$_2$, and KC(CN)$_3$ instead of KSCN (Scheme 1). For **2** yield: 36%. Anal. calc. for $CoC_{15}H_{21}N_9O_3Se_2$ (592.25): C, 30.42; H, 3.57; N, 21.29%. Found: C, 30.73; H, 3.49; N, 21.78%. Characteristic IR data (cm$^{-1}$): $\nu$(N–H) = 3167, 3328; $\nu$(C≡N) = 2087; $\nu$(C=N) = 1661. For **3** yield: 73%. Anal. calc. for $CoC_{15}H_{19}N_{13}O_4$ (504.34): C, 35.72; H, 3.80; N, 36.10%. Found: C, 35.93; H, 3.99; N, 35.87%. Characteristic IR data (cm$^{-1}$): $\nu$(N–H) = 3182, 3311; $\nu$(C≡N) = 2160; $\nu$(C=N) = 1668. For **4** yield: 66%. Anal. calc. for $CoC_{15}H_{19}N_{11}O_{7.16}$ (526.92): C, 34.16; H, 3.61; N, 29.23%. Found: C, 34.63; H, 3.45; N, 29.81%. Characteristic IR data (cm$^{-1}$): $\nu$(N–H) = 3176, 3305; $\nu$(C≡N) = 2185; $\nu$(C=N) = 1663.

{[Co(H$_2$dapsc)(H$_2$O)(N$_3$)][Co(H$_2$dapsc)(N$_3$)$_2$]N$_3$}·4H$_2$O (**5**). *Method B.* The ligand dapsc (2,6-diacetylpyridinebis(semicarazone)) (148 mg, 0.5 mmol) was suspended in ethanol-water mixture (4:1, 20 mL) at 55°C. Then a solutions of cobalt(II) nitrate hexahydrate (146 mg, 0.5 mmol) in 4 mL water and NaN$_3$ (65 mg, 1 mmol) in the same amount of water were added to the ligand suspension. The resulting clear solution was filtered and cooled. The solution was slowly evaporated at 20°C for one to three days. The precipitated red brown crystals were filtered off, washed with a small amount of water, and dried in vacuo. For **5** yield: 53%. Anal. calc. for $Co_2C_{22}H_{40}N_{26}O_9$ (930.60): C, 28.39; H, 4.33; N, 39.13%. Found: C, 28.53; H, 4.68; N, 38.41%. Characteristic IR data (cm$^{-1}$): $\nu$(N–H) = 3174, 3305; $\nu$(–N=N$^+$=N$^-$) = 2017, 2046; $\nu$ (C=N) = 1662.

### 3.2. X-ray Crystal Structure

X-ray data for a single crystals of **1–5** were collected on a *CCD* diffractometer Agilent XCalibur with EOS detector (Agilent Technologies UK Ltd., Yarnton, Oxfordshire, UK) using graphite-monochromated MoK$_\alpha$ radiation ($\lambda$ = 0.71073 Å) and treated by CrysAlisPro software for cell refinement, data collection, and data reduction with empirical absorption correction (Scale3AbsPack) of the experimental intensities [55]. The structure was solved by direct methods and refined against all $F^2$ data (SHELXTL) [56]. All non-hydrogen atoms were refined with anisotropic thermal parameters, positions of hydrogen atoms were obtained from difference Fourier syntheses and refined with riding model constraints. The X-ray crystal structures data have been deposited with the Cambridge Crystallographic Data Center, with reference codes *CCDC* 1952369- 1952373. Selected crystallographic parameters and the data collection and refinement statistics are given in Table S11.

### 3.3. Physical Measurements

The C, H, and N elemental analyses were carried out with a vario *MICRO* cube analyzing device. IR spectra were recorded on a Vertex 70 V instrument in the range from 600 to 4000 cm$^{-1}$ using polycrystalline samples. Absorption bands in the IR spectra were assigned on the basis of literature data. Static magnetic properties measurements (*DC* magnetization) were performed using vibrating sample magnetometer (*VSM*) installed to physical properties measurements system (*PPMS-9*, *Quantum Design*). Dynamic magnetic properties measurements (*AC* magnetization) were performed using *AC* Measurement System (*ACMS*) installed to the *PPMS-9* set-up.

The sample in polycrystalline (powder) form was loaded into a gelatin capsule and glued to the standard sample holder. In order to get magnetic properties of the metal center diamagnetic contribution of sample holder and the ligand was subtracted. For this purpose, the sample holder with the capsule were measured independently. The diamagnetic contribution from ligand was calculated using Pascal's constants.

### 3.4. Computational Calculations

Quantum chemical calculations of the *ZFS* (*D*-tensor) and *g*-tensor parameters for all complexes were performed with post-Hartree-Fock multireference wavefunction (*WF*) approach based on the state

averaged complete active space self-consistent field calculations (*SA-CASSCF*) [57–59] complemented by the N-electron valence second-order perturbation theory (*NEVPT2*) [60–63]. In the state-averaged approach, all multiplets for a given electron configuration were equally weighted. Scalar relativistic effects were taken into account by a standard second-order Douglas-Kroll-Hess (*DKH*) procedure [64]. For calculations a segmented all-electron relativistically contracted version [65] of Ahlrichs polarized triple-ζ basis set *def2-TZVP* [66–68] was used for all atoms. Dominant spin–orbit coupling contributions from excited states were calculated through quasi-degenerate perturbation theory (*QDPT*) [69], in which an approximation to the Breit–Pauli form of the spin–orbit coupling operator (*SOMF* approximation) [70] and the effective Hamiltonian theory [71] were utilized. The *CASSCF* active space was constructed from five MOs with predominant contribution of 3*d*-AOs and seven electrons, corresponding to Co(II) ion—*CAS* (7.5). All possible multiplet states arising from the d7 configuration were included into WF expansion – 10 quartet ($S = 3/2$) and 40 doublet ($S = 1/2$) states. The ab initio ligand field theory [72,73] analysis was done for *CAS* (7.5) calculations. Atomic coordinates have been taken from the single crystal X-ray diffraction data. In selected inconsistent structures, positions of hydrogen atoms were optimized employing density functional theory with BP86 functional and Ahlrichs polarized basis set *def2-TZVP*. Molecular frame of axes has been chosen in such a way that *Z* axis goes along Co ion and donor atoms of axial ligands, *X* axis lies in the plane of the $H_2$dapsc ligand, while *Y* axis is orthogonal to it. All calculations were done by the *ORCA* program (*ver. 4.0.1.2*) [74,75].

## 4. Conclusions

The five new Co(II) heptacoordinate complexes with the equatorial 2,6-diacetylpyridine bis(semicarbazone) ligand ($H_2$dapsc) and various axial pseudohalide ligands were synthesized. The complexes reveal distorted pentagonal bipyramidal geometry which results from pentacoordination of the nearly planar $H_2$dapsc ($N_3O_2$), and two the same or different apical ligands (SCN⁻, SeCN⁻, [N(CN)$_2$]⁻, [C(CN)$_3$]⁻, N$_3$⁻, and $H_2$O) perpendicular to the equatorial pentagon plane. In the case the N$_3$⁻ apical ligand, the crystals of **5** contain two different complexes in the same lattice linked by hydrogen bonds: Neutral [Co($H_2$dapsc)(N$_3$)$_2$] and cationic [Co($H_2$dapsc)(N$_3$)($H_2$O)]⁺. The theoretical analysis of *DC* susceptibility exhibited the positive magnetic anisotropy (*D*) for all complexes with non-zero rhombicity parameter. The calculated values of magnetic anisotropy are in good agreement with the experimental values of *D* (Table 1), which for all complexes under consideration are close to 35−40 cm⁻¹, indicating a weak effect of the nature of axial ligands on the magnitude of magnetic anisotropy parameter for the seven-coordinate mononuclear Co(II) complexes with easy-plane magnetic anisotropy. It is well established that the large positive *D* parameter in pentagonal bipyramidal Co(II) complexes is a result of the spin–orbit mixing of the ground quartet state with excited states, two being quartets and the others are doublets, Table 3 [20,33,34]. An analysis of the literature data on the influence of the coordination environment on the *ZFS* parameter in the Co(II) seven-coordinated complexes shows that the *D* value increases in the case of weaker axially coordinated σ-ligands and a more symmetric equatorial ligand [20,33,34]. To increase the positive *D* value, the energy difference between the ground and quartet excited states should be reduced, which will lead to an increase of the interaction between them. The introduction of weak σ-donors into the apical positions of the complexes will reduce the energy difference between these orbitals and thereby enhance the coupling and increase the positive *D* value. Table S12 presents the *D* parameters of the synthesized **1–7** complexes compared to the *D* parameters of previously reported mononuclear pentagonal-bipyramidal Co(II) complexes with the 2,6-diacetylpyridine-based opened acyclic ligands. The Co(II) complexes with the equatorial $H_4$daps ligand (**13–15**, Table S12) clearly demonstrate the effect of the nature of axial ligands on the *D* value [34]. Among them, complex **13** has the highest positive *D* value among the seven-coordination cobalt systems and contains two weak σ-donor ligands (MeOH) in the axial positions. Full or partial replacement of these donors by stronger σ-ligands (NCS⁻) leads to a decrease of *D* value. Along with spin-orbit coupling of the ground state with an excited quartet states, mixing with doublet states also has a significant effect on the parameter of easy-plane magnetic anisotropy in the pentagonal bipyramidal

Co(II) complexes (Table 3). Note in this connection that in the complex **13**, the equatorial coordination environment is more symmetrical compared to the complexes **14, 15**, since unlike the latters, complex **13** contains an equatorial ligand in a dianionic conjugated form ($H_2$daps), which increases the positive contribution to *D* parameter [34]. As for the complexes **1**–**7** (Table S12) synthesized by us, the high positive *D* value for complex **6** compared to complexes **3**, **4**, and **7** is the result of the presence of weak donor ligands ($H_2O$) in the axial positions of **6**. The experimental *D* values for isostructural complexes **1** and **2** with axial ligands $NCS^-$ and $NCSe^-$ are somewhat different (Table S12), while the theoretical *D* values for these complexes are almost identical (38.02 and 37.73 $cm^{-1}$, respectively, Table 1) and actually coincide with the *D* value for complex **15** with a neutral equatorial ligand $H_4$daps and two axial ligands $NCS^-$(Table S12). Relatively high *D* value for the azide complex **5** (Table S12), which has an unusual structure containing two different complexes (neutral and cationic) in the same crystal lattice, possibly related to a more symmetric coordination environment in the azide complexes compared to complexes **1** and **2**. Coordination polyhedra in **5** are less distorted than in latters.

The complexes **1**–**7** demonstrate the slow magnetic relaxation in weak *DC* field, i.e., are field-induced *SIMs*. For all complexes, the relaxation is well described in the whole temperature range by combination of Raman and direct spin–phonon processes (Table 4).

**Supplementary Materials:** The following are available online at http://www.mdpi.com/2312-7481/5/4/58/s1, Table S1. Shape analysis for the metal centers of complexes **1**–**7**; Table S2. Selected bond lengths (Å) and angles (°) in coordination polyhedra of **1**–**7**; **Figure S1**. Fragments of crystal structures of **1** (a), **2** (b), **3** (c), **4** (d), **6** (e) and **7** (f); Table S3. Geometric parameters of H-bonds in crystal structures **1**–**5**; Figure S2. Fragments of **5** crystal structure: *ac* projection (a) and *ab* projection (b); Figure S3. Stacking interactions of tcm ligands of adjacent molecules in the crystal structure of **4**; Figure S4 The *DC* field-dependencies of *AC* susceptibility ($\chi''$) at 2.0 K and two frequencies (100 Hz, 1000 Hz) for **1** and **5**; Figure S5 Argand (Cole−Cole) plots from 2.0 to 7 K and Arrhenius plots of relaxation times as $\ln(\tau)$ versus 1/T under DC fields for **1** (a), **2**(b), **3** (c), **4** (d), **5** (e), **6** (f) and **7** (g); Tables S4–S10. Best fit parameters of the generalized Debye model for the Cole-Cole plot of complexes **1**–**7** under *DC* fields; Table S11. Crystal data and structure refinement for **1**–**5**; **Table S12**. Magnetic anisotropy parameters (*D* parameters) for seven-coordinated Co(II) complexes with the 2,6-diacethylpyridine-based opened acyclic ligands (Figure 1, main text).

**Author Contributions:** V.A.K. and V.D.S. synthesized the complexes and prepared the crystals; D.V.K. solved crystal structures, performed quantum chemical calculations, analyzed *DC* and *AC* magnetic data and wrote the paper together with E.B.Y. and V.A.K.; I.F.G. performed measurements on SQUID magnetometer; E.B.Y. supervised overall work and organized the project.

**Funding:** This work was supported by the Program No. P13 of Presidium of the Russian Academy of Sciences, project No. 0089-2018-1245. The work was done on the topic of the State task (No. 0089-2019-0011) with using of the Analytical Center for Collective Use tool base of the IPCP RAS. D.V.K. acknowledges the Ministry of Science and Higher Education of the Russian Federation (Agreement no. 14.W03.31.0001). The magnetic measurements were carried out at the Federal Center of Shared Facilities of Kazan federal university. The work of I.F.G. was funded by a subsidy allocated to Kazan Federal University for state assignments in the sphere of scientific activities (project No. 3.8138.2017/8.9).

**Conflicts of Interest:** The authors declare no conflict of interest.

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
