# Peer review of "A Series of Field-Induced Single-Ion Magnets Based on the Seven-Coordinate Co(II) Complexes with the Pentadentate (N3O2) H2dapsc Ligand"

_magnetochemistry, doi:10.3390/magnetochemistry5040058_

Round 1

Reviewer 1 Report

This manuscript describes detailed single ion magnetism of mononuclear high spin cobalt(II) complexes with the pentagonal bipyramidal geometry. All experiments are carefully done, which are supported by theoretical calculations. The manuscript is well written and clear. Therefore, the paper would fit very well in Magnetochemistry, as an article, after minor revision.   (1) There is no systematic discussion of magnetostructural correlations or new insights that are found upon a comparison of a number of previously reported similar complexes. Unfortunately, it just seems routine work, and thus largely diminishes the concept or the quality of this work. The systematic discussion by comparison of previous complexes reported for references 17, 19, 20, and 32–34 should be given and summarized in the manuscript.   (2) The authors excluded the possibility to observe the slow magnetic relaxation through Orbach process in all complexes. Usually, the temperature-dependent relaxation time is firstly approximated by an Arrhenius law. Did the authors estimate effective energy barriers for 17? Please calculate the effective energy barrier, experimentally, because the difference between the experimental and calculated effective barrier is really important. This procedure is illustrative of the validity of Raman and Direct processes.   (3) Related to (2), the temperature dependence of relaxation time for 17 was collected under optimum dc field. This fact means that the direct process does not contribute to the relaxation time throughout whole temperature measured. However, the temperature dependence of relaxation time for 17 was fitted using both direct and Raman processes. The authors should explain it reasonably. In addition, Table S5 should be transferred into the main script.   (4) For only two complexes (1 and 7), the fixed value (9) of n power index of Raman process was used without any mention. The authors should explain it reasonably. In addition, Table S11 should be transferred into the main script.

Author Response

To Reviewer #1

Reviewer #1: (1) There is no systematic discussion of magnetostructural correlations or new insights that are found upon a comparison of a number of previously reported similar complexes. Unfortunately, it just seems routine work, and thus largely diminishes the concept or the quality of this work. The systematic discussion by comparison of previous complexes reported for references 17, 19, 20, and 32–34 should be given and summarized in the manuscript. 

Answer: We agree with the reviewer’s remark and made an appropriate addition to the “Conclusion” section of the revised manuscript (Page 12, Lines: 360-396).

Reviewer #1: (2) The authors excluded the possibility to observe the slow magnetic relaxation through Orbach process in all complexes. Usually, the temperature-dependent relaxation time is firstly approximated by an Arrhenius law. Did the authors estimate effective energy barriers for 17? Please calculate the effective energy barrier, experimentally, because the difference between the experimental and calculated effective barrier is really important. This procedure is illustrative of the validity of Raman and Direct processes.   

Answer: The only one possible variant of the description of the relaxation dependence is presented in the work, including a combination of two contributions: Raman and direct processes, which respectively describe the relaxation process in the region of high and low temperatures, respectively. All processes individually and their various combinations (Orbach+direct, Orbach+direct+Raman, Orbach+Raman, direct +Raman) were tested. By the way, an exception to the direct process from consideration does not allow us to describe the experimental data. Adding the Orbach process into the description implies the presence in the systems of a real level corresponding to the spin-inversion barrier. For systems under consideration with a positive parameter D (easy plane magnetic anisotropy), this energetic level is absent [Gomez-Coca, S. etc Nat. Commun. 2014, 5, 1−8.]. Our quantum chemical calculations confirm this statement also.

Reviewer #1: (3) Related to (2), the temperature dependence of relaxation time for 17 was collected under optimum dc field. This fact means that the direct process does not contribute to the relaxation time throughout whole temperature measured. However, the temperature dependence of relaxation time for 17 was fitted using both direct and Raman processes. The authors should explain it reasonably. In addition, Table S5 should be transferred into the main script.  

Answer: It is known [Inorg. Chem. Front., 2017, 4, 1141-1148], there is the balance between the suppression of QTM and the promotion of the direct process. If the QTM is strong, a large dc field is necessary for its efficient suppression and thus an active direct process is promoted eventually. Moreover, all our temperature dependences of relaxation time cannot be described without counting of direct process.

Reviewer #1: (4) For only two complexes (1 and 7), the fixed value (9) of n power index of Raman process was used without any mention. The authors should explain it reasonably. In addition, Table S11 should be transferred into the main script.

Answer: Fixed value n=9 for 1 and 7 have been explained in the revised text. The table was added in main text (Page 9, Line: 257).

Reviewer 2 Report

The manuscript by V.A. Kopotkov et al. entitled: “A series of field-induced single-ion magnets based on the seven-coordinate Co(II) complexes with the pentadentate (N3O2) H2dapsc ligand” reports on the synthesis and crystal structures of five new heptacoordinate Co(II) complexes. The static and dynamic magnetic properties of these new compounds were studied thoroughly by analysis on spin Hamiltonian and theoretical ab initio calculations. Furthermore, the magnetic data were collected also for two related complexes, which were previously reported, but their magnetism was not studied. The authors revealed that all seven compounds behave as field-induced single-ion magnets and their relaxations involve (besides suppressed q. tunneling) direct and Raman processes. The manuscript is written well, the results and their discussion are sound, and I believe that this paper would be interesting to the readers of the Magnetochemistry journal. However, there are a few parts which can be improved.

In my opinion, the biggest trouble of the present manuscript is the refinement of the crystal structures of complexes 4 and 5. First, the authors should refine their structures in current version of SHELXTL as it is recommended by IUCr. If this had been done a referee would be able to judge refinement in a more exact way since res and hkl files would be included in the cif file. Next, the both structures contain water molecules without hydrogen atoms. I understand decision not to model them, when the data are bad and directions of H-atoms are not clear from Fourier maps, but this seems not to be the case now. And even in such bad data scenario it is possible to use dedicated software to find appropriate positions of the H-atoms. Especially when in 4 and 5 the hydrogen bonding directions are rather clear. When inspecting structure of 4 it is apparent that refinement was not done in an optimal way, because the structure contains atoms refined only isotropically and with rather varying values of Uiso. More importantly, the nitrate anion was modelled having disorder over two positions, which is the most probably good approximation, but when inspecting occupation factors one can find that three oxygen atoms nitrate anion from “main” site (occ.f.= 0.924) were refined together with three O atoms from minor site (occ.f.= 0.076) plus extra O1B atom. And this is not good approach. First of all, one can easily see that Uiso values really differ and some atoms have underestimated occ.factors (O1B and O4B) and second, it seems that also N-O distances (minor site) adopt values, which are either too long (> 1.3 A) or too short (1.1 A). Other troubling thing is that the distance between O1B and O4b is very short (1.895 A, they belong to same PART!) and this does not have any physical meaning. I suppose that the authors have two possibilities: a) to use more constraints and restraints (e.g. SADI, EADP)and try to improve disorder model in a way that it provides meaningful result, b) not to model disorder of the nitrate anion (0.076 is really small value of occ.f.) and model only partial occupancy of O1b. To me, it seems more reasonable to use (b), but the authors know the data. The structure of 5 contains again disorder over two positions of the axial ligands and occupation factors were set to 0.5 – in agreement with expected formula. Again, as in the structure of 4 there are isotropically refined atoms which clearly do not have correct occupation factors – especially non-coordinated azide anion (N14 at special position got same occ.f. as N15!). The authors should try to refine structure with other values of occupation factor than 0.5 to get more meaningful results and maybe also anisotropic model. Non-routine part of the crystal structure determination and refinement should be described in experimental chapter. The results of SHAPE calculations should be available to a reader and I suggest providing them in the Supplementary. 5 L173: “This fact and the non-superposition on a single master curve for M vs B plots at 173 different temperatures indicate the presence of considerable magnetic anisotropy in the complexes.” – I assume this is not a good statement since superposition is expected for M vs B/T plots in the case of isotropic system. Typos or correction suggestions:
a) 7. l208: “the correctness of the choice of gx= gy at SQUID magnetometric measurements describing.” – suggestion: “of gx= gy in analysis of magnetic data”.

b) experimental part: Formulas of pseudohalides a provided using square brackets (e.g. K[SCN]), which should be used for coordination compounds and not simple salts.

Therefore, I suggest this article to be published after minor revision

Author Response

Reviewer #2: In my opinion, the biggest trouble of the present manuscript is the refinement of the crystal structures of complexes 4 and 5. First, the authors should refine their structures in current version of SHELXTL as it is recommended by IUCr. If this had been done a referee would be able to judge refinement in a more exact way since res and hkl files would be included in the cif file. Next, the both structures contain water molecules without hydrogen atoms. I understand decision not to model them, when the data are bad and directions of H-atoms are not clear from Fourier maps, but this seems not to be the case now. And even in such bad data scenario it is possible to use dedicated software to find appropriate positions of the H-atoms. Especially when in 4 and 5 the hydrogen bonding directions are rather clear. When inspecting structure of 4 it is apparent that refinement was not done in an optimal way, because the structure contains atoms refined only isotropically and with rather varying values of Uiso. More importantly, the nitrate anion was modelled having disorder over two positions, which is the most probably good approximation, but when inspecting occupation factors one can find that three oxygen atoms nitrate anion from “main” site (occ.f.= 0.924) were refined together with three O atoms from minor site (occ.f.= 0.076) plus extra O1B atom. And this is not good approach. First of all, one can easily see that Uiso values really differ and some atoms have underestimated occ.factors (O1B and O4B) and second, it seems that also N-O distances (minor site) adopt values, which are either too long (> 1.3 A) or too short (1.1 A). Other troubling thing is that the distance between O1B and O4b is very short (1.895 A, they belong to same PART!) and this does not have any physical meaning. I suppose that the authors have two possibilities: a) to use more constraints and restraints (e.g. SADI, EADP)and try to improve disorder model in a way that it provides meaningful result, b) not to model disorder of the nitrate anion (0.076 is really small value of occ.f.) and model only partial occupancy of O1b. To me, it seems more reasonable to use (b), but the authors know the data. The structure of 5 contains again disorder over two positions of the axial ligands and occupation factors were set to 0.5 – in agreement with expected formula. Again, as in the structure of 4 there are isotropically refined atoms which clearly do not have correct occupation factors – especially non-coordinated azide anion (N14 at special position got same occ.f. as N15!). The authors should try to refine structure with other values of occupation factor than 0.5 to get more meaningful results and maybe also anisotropic model. Non-routine part of the crystal structure determination and refinement should be described in experimental chapter.

Answer: Structures of 4 and 5 have been re-refined with taking into account the reviewer 2 remarks. Refinement of structure 4 was performed without counting of nitrate anion disorder over two positions and with O1b=O2W partial occupancy. Isolated azide anion in structure 5 are refined with correct occ.f. In both structures H atoms at solvate water molecules (besides partial occupied) are added in model. The sentences “The NO3- and H2O moieties are disordered over two positions with 0.91 and 0.09 site occupancies” was deleted (Page 3, lines: 100, 101). New figure for complex 4 was added (Page 4, lines 111; 124, 131)

Reviewer #2: The results of SHAPE calculations should be available to a reader and I suggest providing them in the Supplementary.

Answer: SHAPE calculations have been added in SuppInfo (Table S1).

Reviewer #2: 5 L173: “This fact and the non-superposition on a single master curve for M vs B plots at 173 different temperatures indicate the presence of considerable magnetic anisotropy in the complexes.” – I assume this is not a good statement since superposition is expected for M vs B/T plots in the case of isotropic system. Typos or correction suggestions:
a) 7. l208: “the correctness of the choice of gx= gy at SQUID magnetometric measurements describing.” – suggestion: “of gx= gy in analysis of magnetic data”.

b) experimental part: Formulas of pseudohalides a provided using square brackets (e.g. K[SCN]), which should be used for coordination compounds and not simple salts.

Answer: Incorrect sentences have been improved. The sentences “This fact and the non-superposition on a single master curve for M vs B plots …” and “The analysis of the g-tensor components shows…” were modified (Page 5, Line: 175, Page 7, Line: 211). Experimental part was corrected.